# Urban public space initiatives and health in Africa: A mixed-methods systematic review

**Thondoo Meelan**[1‡]*, **Ebele R. I. Mogo**[1‡], **Nnenna Onyemaobi**[2], **Toluwalope Ogunro**[3],
**Damilola Odekunle**[3,4], **Richard Unuigboje**[3,4], **Salimon K. Muyiolu**[5,6],
**Damilola Olalekan**[3,4], **Catherine Dominic**[7], **Abimbola Thomas**[8], **Edwin Ngwa**[9],
**Okwong Walter**[6,10], **Chimba Sanga**[11], **Victor Onifade**[3,4], **Crespo'o Mbe-cho Ndiabamoh**[12],
**Nfondoh Blanche**[9], **Doris Seyinde**[6,13], **Temitope F. Ogunjimi**[6,14], **Clarisse Mapa-
Tassou**[9,15], **Olanike Maria Buraimoh**[6,16], **Stéphanie S. Teguia**[9], **Ghazel Mukhtar**[17],
**Mvendaga P. Iorse**[18], **Colin Farr**[1], **Ayodipupo S. Oguntade**[19], **Ife Olowoniyi**[4],
**Lia Chatzidiakou**[20], **Louise Foley**[1], **Rose Alani**[6,21], **Taibat Lawanson**[3,4], **Felix Assah**[9],
**Tolu Oni**[1]

1 Global Diet and Activity Research Group and Network, MRC Epidemiology Unit, University of Cambridge, United Kingdom, 2 John Snow Inc, 3 Centre for Housing and Sustainable Development, University of Lagos, Lagos, Nigeria, 4 Department of Urban and Regional Planning, University of Lagos, Lagos, Nigeria, 5 Research and Training Institute, Nigerian Meteorological Agency, Oshodi, Lagos, Nigeria, 6 Air Quality Monitoring Research Group, University of Lagos, Lagos, Nigeria, 7 Barts and the London School of Medicine, Queen Mary University of London, London, England, 8 Department of English, University of Lagos, Lagos, Nigeria, 9 Health of Populations in Transition Research Group, University of Yaounde 1, Cameroon, 10 Department of Botany, Faculty of Science, University of Lagos, Akoka, Lagos, Nigeria, 11 Centers for Disease Control and Prevention, Zambia, 12 School of Tropical Medicine and Global Health, Nagasaki University, Nagasaki, Japan, 13 Anchor University, Lagos, Nigeria, 14 Department of Geography, University of Lagos, Lagos, Nigeria, 15 Faculty of Medicine and Pharmaceutical Sciences, University of Dschang, Dschang, Cameroon, 16 Department of Microbiology, University of Lagos, Lagos, Nigeria, 17 Faculty of Medicine, Imperial College London, London, England, 18 Department of Urban and Regional Planning, Yaba College of Technology, Nigeria, 19 Clinical Trial Service Unit & Epidemiological Studies Unit (CTSU), Nuffield Department of Population Health (NDPH), University of Oxford, Oxford, United Kingdom, 20 Yusuf Hamied Department of Chemistry, University of Cambridge, Cambridge, United Kingdom, 21 Department of Chemistry, University of Lagos, Lagos, Nigeria

‡ MT and ERIM are joint senior authors on this work.
* meelan.thondoo@mrc-epid.cam.ac.uk

**Data Availability Statement:** All data and related metadata underlying the findings reported in a

## Abstract

Public space initiatives (PSIs) in African cities can significantly promote health and social well-being, yet their implementation and impact are unknown across the continent. There is a substantial gap in literature on PSIs in African countries, with most studies concentrated in wealthier cities and lacking comprehensive assessments of long-term health impacts. The objective of this study was to synthesise evidence on the typology, location, features, and outcomes of these initiatives as well as the guiding principles that underlie their design and implementation. Employing a mixed-methods model, the study systematically reviews peer-reviewed and grey literature articles, focusing on the types, settings, and outcomes of PSIs. Data is analyzed using the CASP appraisal tool and thematic analysis. We analysed 47 studies, 15 of which were mixed methods, 22 qualitative and 10 quantitative. Sports accounted for 50% of initiatives. 30 of the 47 papers originated from South Africa. Communities viewed initiatives' wellbeing impacts through social, economic, and ecological lenses, with health being but one dimension. The sustainability of initiatives was often limited by

submitted manuscript and attached supplementary files.

**Funding:** This review is part of the ALPhA (Informal Appropriation of public space for Leisure Physical Activity) project funded by the British Academy Urban Infrastructures of Wellbeing Programme (Grant reference UWB190032) awarded to TO. MT, LF, EM, FA, TL, TO are in part funded by the National Institute for Health Research (NIHR) (NIHR 133205) using UK aid from the UK Government to support global health research. The views expressed in this publication are those of the author(s) and not necessarily those of the NIHR or the UK government. TL is also supported by United Kingdom Research and Innovation (UKRI) under the ARUA-UKRI GCRF Partnership Programme for Capacity Building (Ref: ES/T003804/1) which established the African Research Network for Urbanisation and Habitable Cities.

**Competing interests:** The authors have declared that no competing interests exist.

funding, historical marginalization, and competing land uses. Findings underscore the need for more comprehensive, long-term evaluations and cross-sector collaborations to sustain and enhance health-promoting public spaces in African cities.

## 1. Introduction

Africa has one of the fastest rates of urban growth in the world [1]. With unplanned urbanisation, African cities run the risk of getting locked in cycles of informality and poverty where residents face high socio-economic inequalities, lack sufficient resources, and struggle to access basic services [2]. These factors, coupled with unhealthy commodities and ultra-processed foods, are causing cumulative disadvantage in cities by increasing the double burden of infectious and non-communicable diseases (NCDs) [3, 4]. As resources to address health and social impacts of unplanned and inequitable urbanisation are limited, it is crucial to examine and deploy high-impact, city-scale interventions that can rapidly produce health and wellbeing gains.

Public spaces in cities can cut across diverse segments of urban life to create equitable improvements in population health. They can serve as facilitators of health-promoting resources, services, and behaviours [5].Public spaces can also regulate the negative externalities of urbanisation such as pollution [6], injury and obesity. Public spaces are typified by being open, accessible spaces that lie outside the sole control of individuals and in which people engage in individual or group activities [7]. They may include the built environment (e.g., streets, parks, and stadiums), natural spaces (e.g., hills and coasts), blue spaces (e.g., beaches and public swimming pools), and informal spaces (e.g., junctions and spaces under bridges), that the public may appropriate for recreational activities. Urban infrastructure in public spaces play a critical role in how public spaces influence social and behavioural health factors. Adequate design, access, and conditions of urban infrastructure can encourage safe physical activity [8] and social encounters. Adversely, inadequate infrastructure can pose a multitude of risks to health by exposing urban residents to harm such as air pollution, environmental waste, and injury.

Public space initiatives (PSIs) equally promote health and social well-being in urban environments. Despite growing evidence on the health benefits of public spaces, there is a lack of synthesised evidence on health-promoting PSIs in Africa, one of the fastest urbanising regions globally. This gap is particularly pressing as increasing numbers of African urban residents are repurposing poorly designed infrastructure—such as roundabouts, spaces under bridges, and streets—for physical activity and recreation. These non-conducive spaces can expose residents to harmful factors that may outweigh the health benefits of physical activity and hinder the success of PSIs. PSIs typically involve activities like sports, gardening, and leisure use of green spaces, set in urban contexts such as neighborhoods, parks, schools, and sports clubs. The design and implementation of PSIs involve a range of actors including academic institutions, NGOs, government bodies, and community groups, and are driven by principles of health promotion, community participation, and economic development. The outcomes of these initiatives, which span social, economic, educational, and planetary health benefits, are crucial for urban health promotion, with their long-term success dependent on sustained funding, effective land use management, and ongoing community engagement.

This systematic review focused on PSI in Africa with the potential safeguard health through impacting physical activity, diet, or mental health. We examined different characteristics of

these initiatives that can contribute to health, and more specifically to the prevention of NCDs. Our primary objective was to synthesise evidence on the typology, location, features, and outcomes of these initiatives as well as the guiding principles that underlie their design and implementation. We further sought to distil these findings into evidence-informed recommendations on approaches to equitably optimise urban public spaces for health promotion in Africa.

There are several compelling reasons why examining African PSI at a continental level is useful, as opposed to focusing on sub-regions with similar climate or socioeconomic characteristics. Given that the continent is very vast and diverse, we expect potential disparities in the representation and focus of public space initiatives across different African regions. Studying public space initiatives at a continental level enables us to identify these regional disparities and explore the underlying factors contributing to variations in the implementation, overlapping health outcomes and multidimensional impacts of such initiatives. This approach allows for a more comprehensive analysis of the full spectrum of initiatives and to identify innovative approaches and best practices that may not be apparent when focusing on specific sub-regions or countries. This work is relevant for health and non-health city actors and provides knowledge to advance urban health promotion, and inform strategies for designing safe, inclusive, and multi-functional public spaces.

## 2. Methods

This systematic review applied mixed method approaches to integrate qualitative and quantitative findings of relevance to policy and practice [9]. We applied mixed methods in three ways: (i) extraction of qualitative and quantitative findings (ii) analysis of data (iii) synthesis of findings.

### 2.1 Search strategy

The search strategy aimed at identifying qualitative, quantitative, and mixed-methods studies on PSIs to prevent NCDs in African cities. Between May and June 2020, we searched PubMed, Scopus, Web of Science and Global Health databases using predetermined medical subject headings (MeSH) terms and consulted grey literature (S1 Table). In September 2022, we ran a more recent search of the literature and found no additional studies worth adding. We followed PRISMA guidelines [10], complied with PRISMA checklists (S1 and S2 Checklists), quality appraisal checklists (S3 and S4 Checklists) and registered with PROSPERO (CRD42020189285) [11]. We consulted grey literature with a thematic focus on public spaces and/or health in African cities using internet searches (Google and Twitter) and reports from multilateral agencies (WHO, UN-Habitat, UNICEF). To frame and consolidate the search strategy, we consulted local government and non-governmental agents, collaborators at partner universities and topic experts on different intersectoral approaches to non-communicable disease prevention in Africa. We ran a workshop with a wide range of actors from Cape Town, Dakar, Douala, Accra, Lagos, Kampala, Harare and Maputo [3]; they provided insight on how to best frame the search strategy around practice-focused research and recommended additional literature.

### 2.2 Theoretical framework and definitions

The World Health Organization (WHO) and UN-Habitat framework for integrating health into urban and territorial planning [12] informed our study. This framework addresses how to integrate health into public spaces by considering the settings where initiatives are implemented, the principles driving implementation, sectors involved, and entry points for health

creation. We define cities as settlements with a population of at least 50,000 dwellers [13] who live in contiguous dense grid cells that have over 1,500 inhabitants per square kilometre [14]. Finally, we consider PSIs as efforts that create and improve i) physical environments, ii) social environments, and iii) community resources, with the end goal of ensuring the optimal development of cities for urban residents.

## 2.3 Study selection

We used the Covidence review platform (2022) to review, select, and conduct quality assessment of eligible studies. We doubled-screened titles, abstracts, and full-text according to inclusion criteria. We included studies from African Union member states (S3 Table), published between 1990 and 2022, with no language restrictions. The year 1990 was selected as the cutoff point because it marked the beginning of the promotion of the concept of healthy cities [15]. We included studies containing primary or secondary data. We excluded summaries, literature reviews, conference proceedings, commentaries, opinion pieces and narrative overviews that described public space activities but for the scope of this review, did not analyse primary or secondary empirical data. Where conflicts arose in study selection, investigators with recognised experience and expertise on the topic clarified and resolved them. Team members used their language skills to screen and include eligible studies which were not published in English. They then used the list of eligible studies to perform forward and backward reference searches using Google Scholar and a reference list of eligible articles.

**2.3.1 Types of initiatives.** We included initiatives focusing on modifying public spaces in African cities to address NCD risk factors such as physical activity, diet, injury and/or mental health. These included the design, implementation, or maintenance of PSIs to improve (1) physical infrastructure e.g., green space development or (2) social infrastructure in public spaces. We included all study settings and initiatives run by various sectors such as education, environment, or health.

**2.3.2 Types of participants.** There was no limit to the age, gender, or ethnicity of populations targeted in this review.

**2.3.3 Types of outcome measures.** Outcome measures were both quantitative and qualitative. Primary outcome measures included any objective measures of health (e.g., reduction in hypertension prevalence or increase in physical activity) or social outcomes (e.g. improved participation in recreational activities) associated with PSIs. Secondary outcome measures included health behaviours (e.g., walking) and any measured health exposures (e.g., exposure to air pollution). We also documented how studies assessed outcomes, including enabling or limiting factors that could inform future replication.

## 2.4 Data extraction

We used a data extraction template (S1 Text) designed and piloted by two separate senior researchers. For template validation, they each extracted 5% of the articles and compared results. Junior researchers then used the validated template for data extraction from full-text articles. All final articles were doubly extracted, and concordance checked by researchers (S4 Table). Missing data were managed by first contacting study authors for additional information. If the information was not provided, it was documented as unclear. In addition to extracting general information on the countries represented, publication dates, author affiliation and methodological design of the eligible studies, the template served to extract the following data on PSIs:

- The type of activity of PSIs

- The setting where PSIs take place

- The actors involved in the design, funding, and implementation of PSIs

- The underlying principles for designing PSIs

- The motivation for engagement in PSIs

- The outcomes of PSIs and why it is relevant to promote health in urban spaces

- Consideration of long-term maintenance of barriers and facilitators to PSIs

## 2.5 Quality assessment methods

We used standardised templates, a qualitative checklist, and the cohort study checklist of the Critical Appraisal Skills Programme (CASP) for quality assessment of the studies [16] (S3 Checklist and S4 Checklist). The CASP checklist was modified to accommodate cross-sectional studies and captured the following aspects of quantitative studies (i) whether the study asked a clearly focused question, (ii) the suitability of the recruitment of the cohort, (iii) the extent to which bias was measured and minimized, (iv) the consideration given to confounding in the design and analysis of the study, and (v) the implications of the study for practice. It considered the following aspects of qualitative studies (i) the appropriateness of a qualitative approach used, (ii) the appropriateness of the design, recruitment, and data collection strategy, (iii) the consideration given to positionality between the researcher and participants, and (iv) the consideration given to ethical issues, rigor, clarity, and value of the findings.

## 2.6 Analysis and synthesis

The analysis and synthesis plan were informed by the evidence gathered from data extraction. Following iterative discussions with the team members, particularly EM, TO and LF, it was decided that there was insufficient information to conduct a quantitative meta-analysis, given the heterogeneity of study designs, analytic units, and assessment methods. We therefore presented descriptive measures of quantitative studies and selected a thematic synthesis approach to further synthesize findings [17].

We applied a parallel convergent design to the thematic synthesis approach to compare qualitative and quantitative findings concurrently and allowed these findings to simultaneously enrich one another [18]. A thematic synthesis approach is appropriate for synthesising evidence to inform interventions [19–21]. It allows for the integration of mixed methods data into various categories and transformation of data into emergent themes. It can be theory-driven, or in our case, data-driven [20]. In this study, such a method was useful for interpreting specific categories of data, for example, on activities, partners, and motivations, which were then transformed into recommendations for planning and implementing PSIs to prevent NCDs.

The four steps of thematic synthesis were data coding (step 1), the generation of overarching themes (step 2) which were transformed into recommendations (step 3) and validation of findings (step 4) (Table 1). In step 1, we coded the qualitative data into themes [19] and descriptively summarized the quantitative results. We used evidence from step 1 to further identify key themes in step 2. In step 3, we reviewed the overarching qualitative themes and descriptive quantitative output [20] side by side to synthesise them and transform overarching themes into priorities for policy and intervention recommendations. The validation stage 4 consisted of a consultation with stakeholders to review findings, finalise recommendation and set stage for future research.

**Table 1. Summary of the analysis process.**

| The Study Analysis Table | | | | |
|---|---|---|---|---|
| Step | Data | Description | Key questions to explore | Output |
| 1. Data coding | Key extraction domains as per the extraction tool (available in the appendix) CASP tools | Coding of qualitative data Descriptive summary tables of quantitative data | What components, partners, challenges, opportunities and outcomes are associated with PSIs in African cities? What quality issues need to be addressed in future knowledge co-production? | Qualitative codes Descriptive quantitative data |
| 2. Data translation | Quantitative data Qualitative data | Translation of the combined quantitative and qualitative data codes into themes | What components, partners, challenges, opportunities and outcomes are associated with PSIs in African cities? What quality issues need to be addressed in future knowledge co-production? | Overarching themes from the combined qualitative and quantitative data |
| 3. Data synthesis | Quantitative data Qualitative data Grey literature | Transformation of the overarching themes into priorities into policy & intervention recommendations | What are the implications of the research findings for future policy and interventions to address non-communicable diseases via PSIs in Africa? | Provisional recommendations for policy, action and future transdisciplinary research |
| 4. Validation of the emergent priorities | Quantitative data Qualitative data Grey literature | Stakeholder workshop with research steering group members and multisectoral policy and grassroots actors in urban African cities involving: i) Presentation of analytic approach ii) Presentation of the overarching themes identified iii) Presentation of preliminary recommendations iv) Invitation for comments which may agree, disagree, expound on, or add emergent considerations that need to be captured | What are the implications of the research findings for future policy and interventions to address NCDs via PSIs in Africa? | Finalised recommendations for policy, action and future transdisciplinary research |

## 3. Results

Our initial search of databases, grey literature and additional reference searches yielded 33,258 articles (Fig 1). After removing duplicates and screening title/abstracts and full texts, 47 articles were retained (S2 Text). No article from grey literature was retained for systematic analysis because none met our criteria of having primary or secondary data, but we extracted and reviewed 50 grey literature initiatives (S5 Table) to inform policy recommendations (see step 4 of Table 1). We first present general information about the eligible articles (dates, methodological designs, author affiliations), before describing the different types of activity and settings where the initiatives took place, the partners involved in the design, funding and implementation of the initiatives, the underlying principles for designing initiatives, the motivation for engagement, the outcomes of the initiatives. Finally, we report whether there was discussion or consideration of long-term maintenance of barriers and facilitators to public space initiatives favouring health.

Of the 54 African countries, 19 (24%) were represented, with most articles (n = 30 out of 47) originating from South Africa (Table 2). Most articles (n = 46) took place in formal settings in major cities across the continent including Stellenbosch and Johannesburg (South Africa), Bulawayo (Zimbabwe), Kano (Nigeria), Buea (Cameroon), and Bahir Dar (Ethiopia) while only two articles focusing on informal settings (Khayelitsha and Mafalala in South Africa). There was an increase in the number of articles on the topic between 2008 and 2015 (ranging from one to six publications per year). The methodological designs of the articles were qualitative (n = 22), mixed methods (n = 15) and quantitative (n = 10). There was an equal distribution of first and last authors affiliated to institutions in African (seven) and non-African (seven) countries (Table 2). All records screened are reported in S6 Table.

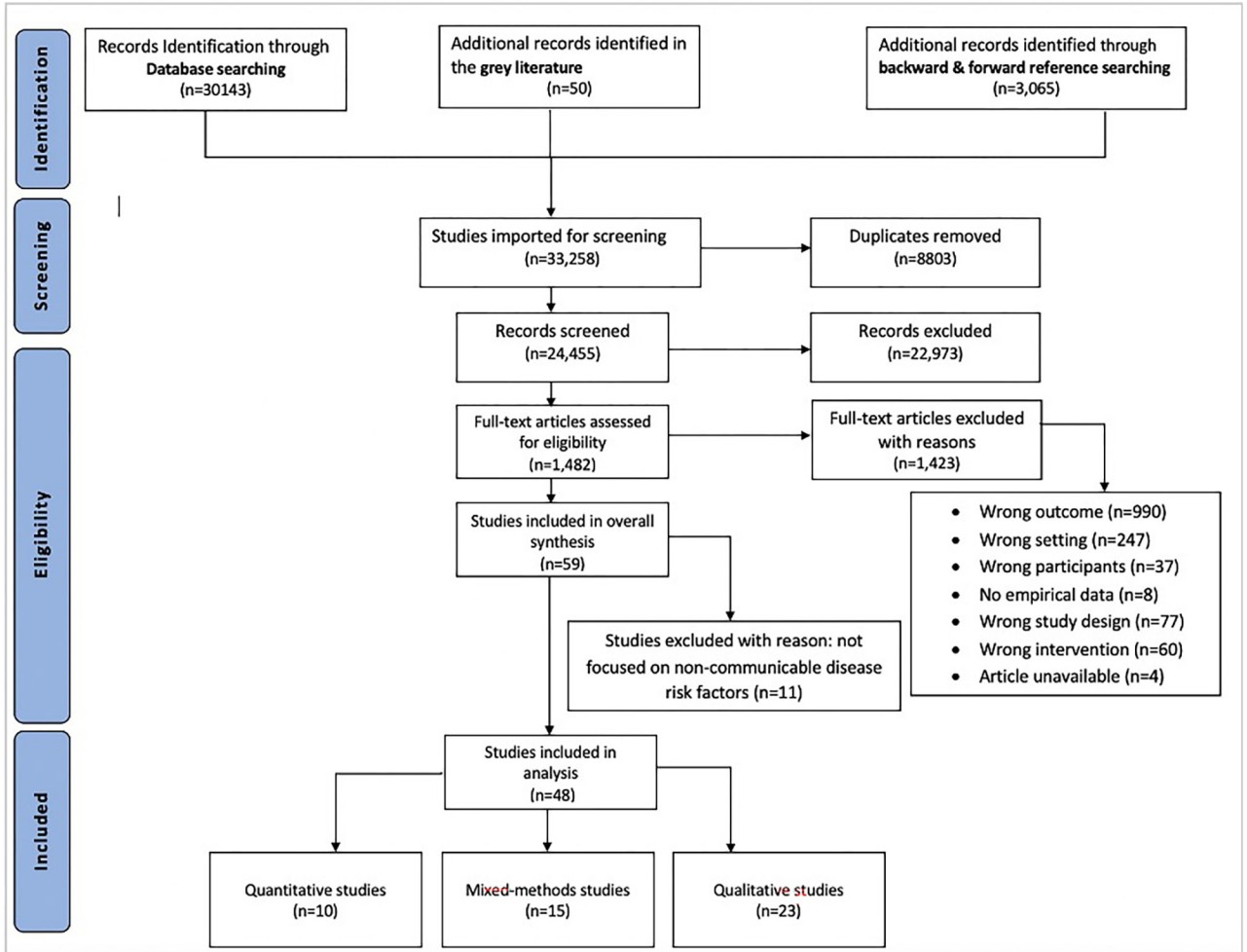

**Fig 1. PRISMA flowchart of study selection showing the different phases of the systematic review process: Identification, screening, eligibility, and inclusion.**

### 3.1 Type of activities and settings

The three different types of activities conducted in PSIs were: sports such as soccer, fitness, and surfing (50%), gardening (35.4%) and leisurely use of green spaces and/or parks (14.6%) (see Table 3). Neighbourhoods (33.3%) were the most common setting for PSIs (neighbourhoods here, are considered as residential spaces, i.e. spaces adjacent to homes), followed by non-residential spaces such as parks and fields (22.9%), schools (16.7%), sports clubs or stadiums (12.5%), tourist sites (4.5%), and the beach (3%). About 40% of all activities were organised in advance with coaches or scheduled in specific locations on set days. Remaining initiatives happened in an ad hoc fashion at different times of the day. Most activities were attended both by groups and individuals (n = 29), the remaining split equally between solo activities (n = 8) and group-based participation only (n = 9).

### 3.2 Partners involved in the design, funding, and implementation of public space initiatives

Design partners included academic organisations (52.1%), research centres (20.8%), non-governmental organisations (12.5%), government (10.4%), international non-profits (8.3%),

**Table 2. Overview of articles.**

| Methodological design | N. of articles | Location | N. of articles |
|---|---|---|---|
| Qualitative | 22 | Formal settings in major cities | 46 |
| Mixed methods | 15 | Informal settlements | 2 |
| Quantitative | 10 | | |
| **Country of implementation** | **N. of articles** | **Year of publication** | **N. of articles** |
| Burkina Faso | 1 | 1991 | 1 |
| Cameroon | 2 | 2000 | 1 |
| England | 1 | 2001 | 1 |
| Ethiopia | 1 | 2002 | 1 |
| Ghana | 2 | 2004 | 1 |
| Kenya | 3 | 2005 | 1 |
| Mauritius | 1 | 2006 | 3 |
| Mozambique | 1 | 2007 | 1 |
| Nigeria | 1 | 2008 | 1 |
| Senegal | 2 | 2009 | 3 |
| South Africa | 30 | 2010 | 6 |
| Zambia | 1 | 2011 | 3 |
| Tunisia | 1 | 2012 | 3 |
| Zimbabwe | 7 | 2013 | 2 |
| Ghana | 2 | 2014 | 1 |
| Uganda | 2 | 2015 | 5 |
| Lesotho | 1 | 2016 | 4 |
| Namibia | 1 | 2017 | 2 |
| Malawi | 1 | 2018 | 3 |
| Botswana | 1 | 2019 | 3 |
| | | 2020 | 2 |
| **Country of first author affiliation** | **N. of articles** | **Country of last author affiliation** | **N. of articles** |
| Cameroon | 1 | Ethiopia | 1 |
| Ethiopia | 1 | France* | 1 |
| France* | 2 | Kenya | 1 |
| Germany* | 1 | Malaysia | 1 |
| Kenya | 1 | Mozambique | 1 |
| Malaysia | 1 | Norway* | 1 |
| Mozambique | 1 | South Africa | 18 |
| Scotland* | 1 | Sweden* | 1 |
| South Africa | 22 | Switzerland* | 1 |
| Switzerland* | 1 | The Netherlands* | 1 |
| The Netherlands* | 1 | United Kingdom* | 1 |
| Tunisia | 1 | United States* | 3 |
| United Kingdom* | 2 | Zimbabwe | 2 |
| United States* | 10 | | |
| Zimbabwe | 4 | | |

schools (x%), and the private sector (2.1%) (Table 4). Funding partners included local universities and/or research institutes (22.9%), international universities and/or research institutes (18.8%), international non-governmental organisations (18.8%), local non-profits and community-based groups (6.25%), and governments (6.25%).

**Table 3. Activity and settings of public space initiatives.**

| Activity types | Percentage | Source(s) |
|---|---|---|
| Sporting activities such as soccer, fitness routines and surfing | 50% | [22–45] |
| Gardening | 35.4% | [46–62] |
| Use of green spaces and/or parks | 14.6% | [13, 30, 63–67] |
| **Settings** | **Percentage** | **Source(s)** |
| Neighbourhood residential compounds | 33.3% | [22, 36, 39, 46, 48, 50, 52–60, 67] |
| Non-residential spaces such as parks and fields | 22.9% | [13, 30, 34, 37, 51, 61–63, 65–66, 68] |
| Sports clubs or stadiums | 12.5% | [27, 35, 40, 42, 43, 64] |
| Tourist sites | 4.5% | [23, 64] |
| Beach | 3% | [22] |

**Table 4. Partners involved in the design and funding of public space initiatives.**

| Design partners | Percentage | Source(s) |
|---|---|---|
| Academic organisations | 52.1% | [22, 29, 30, 33–36, 39–43, 48–50, 52–55, 57, 59, 60, 63, 66] |
| Community based organisations | 16.7% | [27, 37, 53, 56, 58, 59, 62, 63] |
| Non-governmental organisation | 12.5% | [29, 43, 44, 56, 62, 67] |
| Government | 10.4% | [22, 27, 37, 53, 56, 58, 59, 62–64, 66, 67, 69] |
| International non-profits | 8.3% | [27–29, 31, 33, 40, 43, 44, 56, 62, 67] |
| Schools | 4.2% | [29, 33] |
| Private sector | 2.1% | [38] |
| **Funding partners** | **Percentage** | **Source(s)** |
| Local universities and/or research institutes | 22.9% | [34, 42, 45, 47, 48, 56, 58, 59, 61, 63, 67] |
| International universities and/or research institutes | 18.8% | [22, 26, 27, 36, 41, 43, 50, 52, 55] |
| International non-governmental organizations | 18.8% | [28, 29, 31, 33, 40, 44, 55, 62, 64] |
| Local non-profits and community-based groups | 6.25% | [31, 37, 67] |
| Governments | 6.25% | [53, 63, 66] |

## 3.3 The underlying principles for running public space initiatives

Thematic analysis revealed six principles which were used to guide the decision to run initiatives (see Table 5). These included leisure and health promotion (56.25%), participation and the right to the city (47.9%), food security (14.6%), economic development (12.5%), youth development (2.1%), and child-friendly cities (2.1%).

**Table 5. The underlying principles for running public space initiatives.**

| Underlying principles | Percentage | Source(s) |
|---|---|---|
| Leisure and health promotion | 56.25% | [22–29, 31–33, 38, 39, 41–44, 50, 52, 54, 55, 61, 63–65, 68] |
| Participation and the right to the city | 47.9% | [22, 23, 30, 35–38, 42, 43, 45–48, 50, 51, 54, 55, 59, 60, 63, 66] |
| Food security | 14.6% | [57–59] |
| Economic development | 12.5% | [13, 23, 34, 35, 46, 56, 58] |
| Youth development | 2.1% | [27] |
| Child-friendly cities | 2.1% | [67] |

### 3.4 Motivation for engagement

There were eight different motivations for engagement identified across PSIs. The primary motivation for people to engage in PSIs was leisure, play and/or sports (50%) followed by community participation (45.8%) where inclusion, cohesion and governance were promoted. The third most cited motivation (33.3%) was the empowerment of communities through food production and the reduction of food insecurity. Less common motivations included environmental stewardship via waste management or ecological conservation (12.5%); income generation through tourism (6.25%); the promotion of life skills such as self-esteem or youth development (4.2%) and finally the promotion of culture (2.1%).

### 3.5 Outcomes of the initiatives

Based on the six underlying principles used to inform PSIs, a further qualitative analysis led to identifying five outcomes that were considered by the articles: social, economic, human health, education, and planetary health outcomes. Each principle touched at least one outcome but could touch various at a time. For instance, underlying principles such 'leisure and health promotion' were linked to education, health, and social outcomes simultaneously (Fig 2).

There were multiple cases where public space initiatives designed for leisure and health promotions led to multidimensional outcomes on health, education, and social life [27, 46–48]. Examples include initiatives using sports to promote physical activity while educating participants about HIV [28, 68]; the development of palace gardens for ecological, spiritual, social and economic purposes [51]; urban agriculture initiatives to reduce spending on food, improve food security and also promote economic wellbeing [51–53, 60]; walking tours which integrated physical activity, culture and economic development [30, 37]; dance programs which linked family time with physical activity [38, 59]; community gardens which often linked access to food, exercise, and income [41, 60–62] and sports programmes that also built vocational skills [43]. Other examples included articles showing that the rationale to address food security was strongly related to planetary health outcomes [46, 48–50, 52, 54, 55, 57–60, 63].

All studies implicitly reported outcomes, but only three articles explicitly measured and reported health (% healthy food consumption), economic (% of food expenditure) and social outcomes (% of employment). In a study focusing on community gardens in South Africa, 54% of study respondents reported consumption of the healthy food produced (versus none before) and 58% reported healthy lifestyle as an outcome of the intervention [59]. Another

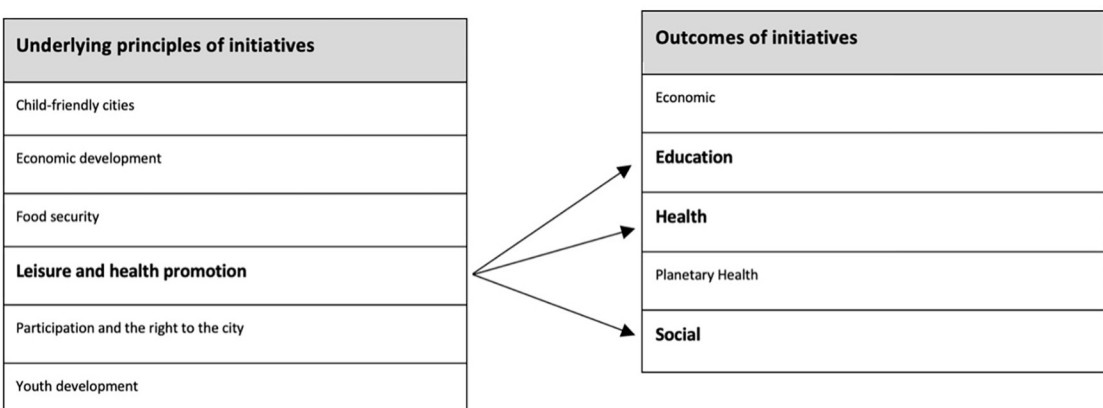

**Fig 2. Links between underlying principles and outcomes pf public space initiatives.**

study reported that total daily food expenditures was reduced by about 20% after a home garden initiative [61]. Finally, a study using soccer and job training to prevent drug abuse and HIV among young men in Cape Town reported that at six months, 28.9% in the intervention groups were employed versus only 9.9% in the delayed intervention group [43].

### 3.6 Consideration of long-term sustenance of public space initiatives

In total, 17 articles raised considerations for sustaining activities in public spaces. Various challenges to long-term planning and sustenance were reported. They included one-time collection of data (therefore limited visibility on long term implications) [43], lack of funding [41, 42, 52, 57] issue of access and competing uses of land [13, 37]. On the other hand, several opportunities for increasing sustenance were reported. They included training and consistent support of permanent staff [39, 62], and allocation of permanent space for the activities [60], keeping inventories of the spaces (e.g., gardens) [59].

### 3.7 Quality assessment results

Data robustness at the quality appraisal stage varied widely due to the heterogeneity of disciplines and studies considered. We did not exclude studies or apply thresholds to them based on quality. Instead, while refining interpretation, we gave more weight to information from more comprehensive studies. Most importantly, we used from the CASP appraisal tool to identify and report quality issues across studies that can serve to inform the design of future initiatives, as done elsewhere by our research team [71]. Twelve studies addressed the CASP appraisal questions more thoroughly and were thus considered more robust than others [30, 36, 37, 50, 52, 54, 55, 66, 72, 73]. Consistent quality issues identified included linguistic and sociocultural biases and their possible impact on data collection and analysis and rigour and bias in the reporting of the data collection and results to enable the separation of the opinion of authors from the findings of the project. There was also a need identified for long-term impact assessment across all studies, to enable an understanding of the true effect of such interventions on health outcomes as well as potential positive and negative externalities.

## 4. Discussion

We analysed 47 studies, 15 of which were mixed methods, 22 qualitative and 10 quantitative. Only 24% of the 54 African countries were represented. Most studies originated from South Africa and sports accounted for 50% of all initiatives. The most common settings for PSIs were neighbourhood plots, parks, and sports clubs. Most initiatives were driven by principles for leisure and health promotion as well as participation and the right to the city. These principles were closely linked and overlapped to different health, economic, social, educational, and planetary health outcomes. Reflecting on these findings, we provide some recommendations in this discussion section by highlighting the implications and evidence gaps for promoting public space initiatives for health in African urban settings.

### 4.1 The added value of this study to scientific literature

Our analysis of peer-reviewed articles reveals a significant gap in the academic and grey literature concerning public space initiatives across most African countries, with a notable concentration of reported initiatives in wealthier African cities. Considering the accumulating evidence demonstrating the health benefits associated with public spaces in urban settings [74, 75], it is imperative for more African nations to adopt and document healthy public space initiatives. Furthermore, our examination of grey literature indicates a rising trend in such

initiatives across various sectors, particularly in transportation and public infrastructure. Examples include car-free days (e.g., Cooperation for Urban Mobility in the Developing World, 2019), open streets [76], play spaces [77], parks [78], public space upgrades [79], and shared road initiatives [80]. Rigorous methodologies for mapping and documenting these initiatives, particularly in the context of mobility within African cities have been advocated for [71]. We echo this call, emphasizing the necessity of empirical assessments to ensure a comprehensive representation of activities continent-wide. Moreover, we emphasize the importance of incorporating historical factors like apartheid into the design of public space initiatives to foster community trust and enable community contributions to new public space developments [81]. A systematic approach to cataloguing public space initiatives in Africa involves identifying existing public spaces, detailing initiatives within these spaces, and establishing research-to-practice partnerships for implementing and evaluating health-promoting activities. The establishment of the African Centre for Public Space could serve as a catalyst in advancing this process further [82].

## 4.2 Reconceptualising public spaces by integrating health and social dimensions

Our study calls for a reconceptualization of public spaces, not only as social spaces but also as crucial health spaces. The findings reveal that a diverse range of activities accommodated by public spaces influences health and social interactions in various ways. Exposure to green spaces, for example, fosters outdoor activities that increase physical activity and social contacts while promoting relaxation[74]. Moreover, public spaces have been found to have greater health benefits for populations with lower socioeconomic status, and perceptions of health benefits vary across different demographic groups and settings. Indeed, public environments such as green spaces have larger health benefits for populations with lower social economic status [83]. Gender can also drive social perceptions on how these spaces affect health and social dimensions of life. And perceptions on health can further vary across settings with an example from Indonesia showing that health-benefits were perceived to be delivered by a combination of factors such as physical activities, recreational activities, and restorative effect of the natural elements, *including* social interaction [84]. Our findings highlight a predominant focus on sports, particularly soccer, in current public space initiatives, which aligns with the primary motivations of individuals to engage in sports, leisure, and play activities (Table 6). Sports serve as a potent driver for harnessing the health benefits associated with public spaces, facilitating physical activity that contributes to improved health outcomes. Additionally, public

Table 6. Motivation to engage in public space initiatives.

| Motivation to engage | Percentage | Source(s) |
|---|---|---|
| Sports, leisure and/or play | 50% | [13, 22, 23, 25–29, 31–33, 35, 36, 38–43, 45, 63, 67, 68, 70] |
| Community participation | 45.8% | [23, 25, 27, 35, 36, 38, 42, 44, 46–48, 50, 53, 55–57, 59, 60, 62, 66, 67] |
| Empowerment of communities through food production and the reduction of food insecurity | 33.3% | [46–50, 52–62] |
| Environmental stewardship via waste management or ecological conservation | 12.5% | [27, 28, 30–32, 43, 51, 58, 63, 65, 66, 68] |
| Income generation through tourism | 6.25% | [33, 35, 64] |
| Promotion of life skills such as self-esteem or youth development | 4.2% | [40, 44] |
| Promotion of culture | 2.1% | [37] |

spaces offer opportunities for community cohesion and participation, which are the second most common motivations for engagement. Therefore, initiatives in public spaces should emphasize the co-benefits of physical activity within settings characterized by strong community participation. As such, community spaces are as much health spaces as they are social spaces.

### 4.3 Work in this area from other regions of the world

Our findings align with research from other regions of the world, indicating that public spaces play a vital role in promoting community health and social well-being. However, disparities in the distribution and utilization of public spaces exist across different demographic groups, influenced by factors such as gender, socioeconomic status, and cultural norms. Understanding these dynamics is essential for designing inclusive and equitable public space initiatives that cater to the diverse needs of urban populations. The involvement of various stakeholders, including government entities, community groups, and international organizations, is critical for funding, designing, and implementing public space initiatives that prioritize health and social outcomes. For instance, collaboration between political actors, community volunteers, grassroots organisations and private sector players were shown to be effective during the Covid-19 pandemic, resulting into improved access to nutrition via public spaces in Kenyan and Nigerian cities [85]. This reflects current evidence from Asia showing that conducting participatory engagement of state and non-state actors and communities in planning and developing public space initiatives is essential to ensure health is a priority on urban agendas [86, 87].

Involving different actors can also avoid action in silos and can encourage cross-cutting partnerships with local communities in designing public space initiatives. Our findings show for example that an initiative integrating sports and HIV testing requires technical input from urban planning and health professionals but also from individuals aware of historical tensions, coach-player relationships, and community needs [37, 61]. Similarly, as most public space initiatives documented were in neighbourhood settings, it is crucial to merge insights from actors who promote spaces for sports and gardening alongside actors who can provide funding, resources, and incentives (for instance decision-makers and developers of residential projects). Evidence from Latin America shows that community-based knowledge is crucial in the planning of projects, the development and interpretation ofinterpreting data to improve and evaluate projects, and building the trust and capacity required to sustain projects [88–91]. Similarly, communities in African cities experience public spaces in different ways which are influenced by gender, socioeconomic status, occupation, age and race [92–96]. This is also true for people's experiences with transport systems [71].Therefore, inviting and integrating communities' subjective experiences in the co-design of initiatives presents untapped potential for health-oriented interventions to respond to social, cultural, and economic needs [11], but can also increase the benefits of initiatives on a larger proportion of urban residents, including vulnerable groups.

Our findings show that communities hold different motivations for engagement in initiatives, with half of the articles reporting primary motivation as leisure, play and/or sports followed by community participation (45.8%). Our findings echo existing literature arguing that motivation can be considered a critical factor in encouraging and maintaining activities that can promote health such as physical activity [94]. As more people are seeking to engage in public space initiatives for leisure, community participation and food security than for environmental stewardship or promotion of culture, it is safe to assume that motivation factors can convey a significant role in designing spaces that can equitably engage different population

groups. Hence, working alongside communities couple provide new opportunities for city actors to increase civic education. Such capacity-building activities will increase demand for health-promoting public spaces and provide space for awareness building and advocacy, through and in collaboration with communities. Finally, health actors should be trained to develop the skills and relationships necessary to engage with public space actors, and vice versa.

## 4.4 Relevance of findings for growth areas and future directions

Despite the increasing recognition of the importance of public space initiatives, several gaps persist in current academic inquiry. Many studies (65%) fail to address considerations for sustaining initiatives on a long-term basis, with challenges such as funding constraints and competing land uses posing significant obstacles. Some studies reported that public space initiatives faced challenges such as lack of funding [41, 42, 52, 57] and competing uses of land [13, 37]. Several suggestions to ensure sustenance of activities were provided, including institutional integration and contextual backing [56] measuring participation, and requesting end-user feedback [33, 49], involvement of non-governmental organisations to continue the activities [29], addressing safety and security concern for sustenance of the spaces [26] and requesting adequate and consistent policies to regulate relevant sectors [23].

Moreover, there is a notable lack of evaluations assessing the health impact of public space initiatives, underscoring the need for more comprehensive and long-term assessments to understand both the immediate and sustained effects on human and planetary health. We call for a more complex evaluation of initiatives in rapidly urbanising cities, noting their immediate and sustained impact on human and planetary health [97]. In the context of public space, longer-term evaluations will create a clearer picture of potential short and long-term benefits as well as the unintended consequences of public space initiatives. Evaluation tools and frameworks offer valuable resources for assessing the multifaceted impacts of public space initiatives, and understanding the extent to which global development agendas, particularly around urban infrastructure provision, integrate health into urban dimensions. For example, whole-of-society frameworks such as the community wellbeing framework [98],can provide for a robust evaluation and understanding of the impact of public space initiatives on health. Other tools include the UN-Habitat public space assessment toolkit and International Guidelines on Urban and Territorial Planning [99]. Legal tools to enforce health considerations in public space development can also be considered as rights-based approaches can inform existing decision-making on public spaces, to ensure that they are health-promoting and are appropriated safely by individuals and communities. In contrast to punitive approaches, rights-based approaches can lead to a greater consideration of what communities need and expect from public spaces, including but not limited to safety, community, sports, and security.

Incorporating health impact assessments (HIAs) into urban development projects can help ensure that public spaces are safe, inclusive, and conducive to promoting health and wellbeing for all urban residents. HIAs can strengthen the current knowledge base to understand the effects of public space initiatives on health behaviours and outcomes but can also help identify non-health externalities related to hazards such as climate change. Given the competing uses of land, HIAs of public spaces can be useful to further assess how the effects of initiatives on health are distributed across population groups. Finally, HIAs provide opportunities to foster interdisciplinary collaborations among health, urban planning, sports, and policy experts is essential for designing and evaluating formal and informal public space initiatives that address the diverse needs of urban populations while promoting health equity and sustainability.

### 4.5 Strengths and limitations

To our knowledge, this is the first systematic review to consider how public space initiatives can be optimised for health promotion in Africa. The purpose of the paper was not to apply insights but to understand the knowledge landscape of initiatives on this topic on the continent. We consider the findings noteworthy as it highlights the need for more data from different countries on the continent to build more contextual evidence for countries across geographic, climatic, and socioeconomic contexts. We considered various forms of literature to examine the features and implications of such initiatives and interpreted our findings towards actionable recommendations. For the systematic review, we accorded primacy to peer-reviewed literature and therefore, may not have captured the breadth of ongoing public space initiatives in cities on the continent. There is also a risk of having missed initiatives that were implemented but not documented. We addressed these limitations by conducting consultations and reviewing grey literature to identify initiatives occurring within informal systems such as cultural meetings and spaces that may not be formally documented or published. Nonetheless, we recognise that there are likely to be other public space initiatives that have not been documented in the public domain. An added value lies in the diversity of the study team, working from various African countries including but not limited to South Africa, Nigeria, Kenya, and Cameroon. The recommendations were also refined through close engagements activities with local stakeholders from the public sector, civil society, academia, and professional associations to contextualise and operationalise findings as much as possible. Finally, although data robustness at the quality appraisal stage varied widely due to the heterogeneity of disciplines and studies considered, the use of the CASP appraisal tool enabled the study team to weigh the quality of the studies and provide the best evidence to design future initiatives.

## 5. Conclusion

Given ongoing urban infrastructure expansion on the continent, public space initiatives are a high-impact avenue to equitably improve the health of the public. There is a need for more comprehensive documentation and implementation of interventions that address public health through initiatives in public spaces across the continent. Existing evidence on such interventions shows that use of public spaces for sporting activities and gardening are dominant, and that public space initiatives are valued for their social, economic, ecological and health benefits.

Scientific enquiry is needed on how public health practitioners in African cities collaborate with urban development and community actors to incorporate health into the planning of public space initiatives. Research on the political dimension of public space initiatives can uncover how actors navigate land-use priorities, insecure tenure, and conditions of urban precarity. In turn, this can unravel strategies to ensure the long-term sustainability of health-promoting public space initiatives and support actors in designing spaces that can reduce poverty, increase social cohesion, and promote ecological restoration while supporting health. Finally, further research should investigate the extent to which community preferences are considered in designing and implementing policies, particularly in informal systems where funding and resource mobilisation approaches remain unexamined.

## Supporting information

**S1 Checklist. PRISMA checklist.**
(DOCX)

**S2 Checklist. PRISMA abstracts checklist.**
(DOCX)

**S3 Checklist. CASP qualitative checklist (for qualitative studies).**
(DOCX)

**S4 Checklist. Modified CASP cohort study checklist (for quantitative studies).**
(DOCX)

**S1 Table. Search strategy and terms.**
(DOCX)

**S2 Table. Systematic review inclusion and exclusion criteria.**
(DOCX)

**S3 Table. List of included countries.**
(DOCX)

**S4 Table. List of senior and junior researchers.**
(DOCX)

**S5 Table. Characteristics of grey literature studies.**
(DOCX)

**S6 Table. All records screened.**
(XLSX)

**S1 Text. Data extraction template.**
(DOCX)

**S2 Text. Characteristics of included studies.**
(DOCX)

**S1 File. All data extracted from each study.**
(XLSX)

**S2 File. Quality assessment results for each study.**
(XLSX)

**S1 Fig. Prisma flowchart.**
(PDF)

**S2 Fig. Links between underlying principles and outcomes pf public space initiatives.**
(TIF)

## Author Contributions

**Conceptualization:** Thondoo Meelan, Ebele R. I. Mogo, Doris Seyinde, Colin Farr, Louise Foley, Taibat Lawanson, Felix Assah, Tolu Oni.

**Data curation:** Ebele R. I. Mogo, Nnenna Onyemaobi, Toluwalope Ogunro, Damilola Odekunle, Richard Unuigboje, Salimon K. Muyiolu, Damilola Olalekan, Catherine Dominic, Abimbola Thomas, Edwin Ngwa, Okwong Walter, Chimba Sanga, Victor Onifade, Nfondoh Blanche, Temitope F. Ogunjimi, Clarisse Mapa-Tassou, Stéphanie S. Teguia, Ghazel Mukhtar, Mvendaga P. Iorse, Ayodipupo S. Oguntade, Ife Olowoniyi, Lia Chatzidiakou, Louise Foley, Tolu Oni.

**Formal analysis:** Thondoo Meelan, Ebele R. I. Mogo, Nnenna Onyemaobi, Damilola Olalekan, Edwin Ngwa, Crespo'o Mbe-cho Ndiabamoh, Olanike Maria Buraimoh, Stéphanie S. Teguia, Colin Farr, Louise Foley, Rose Alani, Taibat Lawanson, Felix Assah.

**Funding acquisition:** Tolu Oni.

**Investigation:** Ebele R. I. Mogo, Nnenna Onyemaobi, Crespo'o Mbe-cho Ndiabamoh, Taibat Lawanson, Tolu Oni.

**Methodology:** Thondoo Meelan, Ebele R. I. Mogo, Nnenna Onyemaobi, Crespo'o Mbe-cho Ndiabamoh, Olanike Maria Buraimoh, Stéphanie S. Teguia, Ghazel Mukhtar, Mvendaga P. Iorse, Colin Farr, Rose Alani, Taibat Lawanson, Felix Assah, Tolu Oni.

**Project administration:** Ebele R. I. Mogo, Tolu Oni.

**Resources:** Ebele R. I. Mogo, Tolu Oni.

**Supervision:** Ebele R. I. Mogo, Taibat Lawanson, Tolu Oni.

**Validation:** Thondoo Meelan, Ebele R. I. Mogo, Taibat Lawanson, Tolu Oni.

**Visualization:** Ebele R. I. Mogo, Taibat Lawanson, Tolu Oni.

**Writing – original draft:** Thondoo Meelan, Ebele R. I. Mogo, Edwin Ngwa, Taibat Lawanson, Tolu Oni.

**Writing – review & editing:** Thondoo Meelan, Ebele R. I. Mogo, Toluwalope Ogunro, Damilola Odekunle, Richard Unuigboje, Salimon K. Muyiolu, Damilola Olalekan, Catherine Dominic, Abimbola Thomas, Edwin Ngwa, Okwong Walter, Chimba Sanga, Victor Onifade, Crespo'o Mbe-cho Ndiabamoh, Nfondoh Blanche, Doris Seyinde, Temitope F. Ogunjimi, Clarisse Mapa-Tassou, Stéphanie S. Teguia, Ghazel Mukhtar, Mvendaga P. Iorse, Colin Farr, Ayodipupo S. Oguntade, Ife Olowoniyi, Lia Chatzidiakou, Louise Foley, Rose Alani, Taibat Lawanson, Felix Assah, Tolu Oni.

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
