## [Decision Letter · Decision Letter 0]

30 Jan 2024

PGPH-D-23-01544

Urban Public Space Initiatives and Health in Africa: A Mixed-Methods Systematic Review

Dear Dr. Meelan,

Thank you for submitting your manuscript to PLOS Global Public Health. After careful consideration, we feel that it has merit but does not fully meet PLOS Global Public Health’s publication criteria as it currently stands. Therefore, we invite you to submit a revised version of the manuscript that addresses the points raised during the review process.

Please note that we have only been able to secure a single reviewer to assess your manuscript. We are issuing a decision on your manuscript at this point to prevent further delays in the evaluation of your manuscript. Please be aware that the editor who handles your revised manuscript might find it necessary to invite additional reviewers to assess this work once the revised manuscript is submitted. However, we will aim to proceed on the basis of this single review if possible. 

The reviewer has raised a number of concerns that need attention. Specifically, have provided suggestions to restructure and focus the discussion section, and to double check the spelling in the manuscript. 

Could you please revise the manuscript to carefully address the concerns raised?

We look forward to receiving your revised manuscript.

Kind regards,

Johanna Pruller, Ph.D.

PLOS Staff Editor

Journal Requirements:

1. Please provide separate figure files in .tif or .eps format only and remove any figures embedded in your manuscript file. Please also ensure all files are under our size limit of 10MB.

2. We have noticed that you have uploaded Supporting Information files, but you have not included a list of legends. Please add a full list of legends for your Supporting Information files after the references list. 

3. In the online submission form, you indicated that "The data that was extracted from available published and peer-reviewed studies that support the findings of this study are available from the corresponding author, [MT], upon reasonable request". All PLOS journals now require all data underlying the findings described in their manuscript to be freely available to other researchers, either 1. In a public repository, 2. Within the manuscript itself, or 3. Uploaded as supplementary information.

Additional Editor Comments (if provided):

Reviewers' comments:

Reviewer's Responses to Questions

**Comments to the Author**

1. Does this manuscript meet PLOS Global Public Health’s publication criteria? Is the manuscript technically sound, and do the data support the conclusions? The manuscript must describe methodologically and ethically rigorous research with conclusions that are appropriately drawn based on the data presented.

Reviewer #1: Yes

2. Has the statistical analysis been performed appropriately and rigorously?

Reviewer #1: N/A

3. Have the authors made all data underlying the findings in their manuscript fully available (please refer to the Data Availability Statement at the start of the manuscript PDF file)?

Reviewer #1: Yes

4. Is the manuscript presented in an intelligible fashion and written in standard English?

Reviewer #1: Yes

5. Review Comments to the Author

Reviewer #1: The authors conducted a review of evidence on the typology, location, features & outcomes on public space initiatives in Africa from a safeguarding health lens. Thank you for the opportunity to review this manuscript.

I would like to commend the authors for conducting such a thorough review. See my comments below.

I understand that a lot of global health work focuses on Africa broadly, but the continent is vast a very diverse. The authors should make a stronger case for why studying african public space initiatives as a whole, as opposed to say, sub-regions with similar climate and climate-change related challenge, or say, a subset of countries with similar socioeconomics, as they may be similar in terms of advances in urban and rural infrastructure. As the authors note, the vast majority of papers they identified are from South Africa. Both in terms of climate (types of seasons etc) and socioeconomics, South Africa is closer to many Western countries than many of its African counterparts. Perhaps it makes more sense to split this paper. otherwise, can you apply to sub-saharan/central/west African countries the insights gleaned from a number of papers with a vast majority about South Africa?

Double check spelling/numbers. in section 3.2, the number after school is “x.”

I appreciate the robustness of the methods, and the thoroughness with which the authors wrote the results section.

The discussion section is very long.

I suggest that the authors limit the discussion to:

1. hilighliting key findings

2. What do findings uniquely add to the literature?

3. explain/theorize about your findings

4. put findings in concert with existing literature, in this case, perhaps work in this area based on other regions of the world.

5. Highlight gaps in the literature, growth areas, future directions.

6. Highlight study limitations

In addition to community cohesion and physical activity, I recommend that authors discuss nuances between different types of nature public spaces, and their respective benefits to the environment especially in the context of climate change and also how such public spaces can serve as an opportunity for simple relaxation, which in itself can be good for health.

The recommendations to governments, while very good, belong in a separate policy brief or perspective. I think they go beyond the stated goal/scope of the manuscript.

Thank you for the opportunity to review this manuscript.

6. PLOS authors have the option to publish the peer review history of their article (what does this mean?). If published, this will include your full peer review and any attached files.

**Do you want your identity to be public for this peer review?** For information about this choice, including consent withdrawal, please see our Privacy Policy.

Reviewer #1: No

---

## [Decision Letter · Decision Letter 1]

19 Jun 2024

PGPH-D-23-01544R1

Urban Public Space Initiatives and Health in Africa: A Mixed-Methods Systematic Review

Dear Dr. Meelan,

Thank you for submitting your manuscript to PLOS Global Public Health. After careful consideration, we feel that it has merit but does not fully meet PLOS Global Public Health’s publication criteria as it currently stands. Therefore, we invite you to submit a revised version of the manuscript that addresses the points raised during the review process.

The manuscript has been evaluated by two reviewers, and their comments are available below.

One reviewer has recommended Accept, while the other has raised a few minor concerns. Specifically, they suggest that you increase the amount of information delivered in the Abstract and Introduction section of your manuscript.

Could you please carefully revise the manuscript to address all comments raised?==============================

We look forward to receiving your revised manuscript.

Kind regards,

Johanna Pruller, Ph.D.

PLOS Staff Editor

Journal Requirements:

Additional Editor Comments (if provided):

Reviewers' comments:

Reviewer's Responses to Questions

**Comments to the Author**

1. If the authors have adequately addressed your comments raised in a previous round of review and you feel that this manuscript is now acceptable for publication, you may indicate that here to bypass the “Comments to the Author” section, enter your conflict of interest statement in the “Confidential to Editor” section, and submit your "Accept" recommendation.

Reviewer #1: All comments have been addressed

Reviewer #2: (No Response)

2. Does this manuscript meet PLOS Global Public Health’s publication criteria? Is the manuscript technically sound, and do the data support the conclusions? The manuscript must describe methodologically and ethically rigorous research with conclusions that are appropriately drawn based on the data presented.

Reviewer #1: Yes

Reviewer #2: Yes

3. Has the statistical analysis been performed appropriately and rigorously?

Reviewer #1: N/A

Reviewer #2: Yes

4. Have the authors made all data underlying the findings in their manuscript fully available (please refer to the Data Availability Statement at the start of the manuscript PDF file)?

Reviewer #1: Yes

Reviewer #2: Yes

5. Is the manuscript presented in an intelligible fashion and written in standard English?

Reviewer #1: Yes

Reviewer #2: Yes

6. Review Comments to the Author

Reviewer #1: Thank you for the opportunity to review this iteration.

The authors have addressed all my comments adequately.

Reviewer #2: Review PGPH-D-23-01544

Thanks for inviting me to evaluate this interesting review paper by Thondoo et al. Based on the existing studies, it provides a good overview of public space initiatives (PSI) in African contexts. The method is clear, the results are reasonable, and the study is well-written. I see some good value in publishing such works; thus, I am mostly positive about their publication. I only have a few minor points, which might be better clarified.

• The abstract is too brief to say anything clearly. I am not sure if it is a journal requirement. But if it is not, then I suggest expanding the abstract a bit more to give further information to the reader, the background of the paper, and some additional info about the PSI-related activity type and settings.

• The intro could give a bit more information on PSI interventions from a theoretical viewpoint, such as why and what types are common in contexts of Africa. I missed some points about the PSI in the intro; after reading the result, I got the idea clearly, but the authors could add a few more lines about the PSI in the intro, such as Activity types and Underlying principles.

• As a result, table 2 can be converted into charts and maps that would be easy to follow.

Other parts of the paper are quite clear, and I do not have further comments.

Good luck modifying the paper. I am looking forward to the final version.

Best Wishes.

Labib

7. PLOS authors have the option to publish the peer review history of their article (what does this mean?). If published, this will include your full peer review and any attached files.

**Do you want your identity to be public for this peer review?** For information about this choice, including consent withdrawal, please see our Privacy Policy.

Reviewer #1: No

Reviewer #2: **Yes: **S.M. Labib

---

## [Decision Letter · Decision Letter 2]

23 Aug 2024

Urban Public Space Initiatives and Health in Africa: A Mixed-Methods Systematic Review

PGPH-D-23-01544R2

Dear Dr. Meelan,

We are pleased to inform you that your manuscript 'Urban Public Space Initiatives and Health in Africa: A Mixed-Methods Systematic Review' has been provisionally accepted for publication in PLOS Global Public Health.

Best regards,

Ryan Essex

Academic Editor

Reviewer Comments (if any, and for reference):

Reviewer's Responses to Questions

**Comments to the Author**

1. If the authors have adequately addressed your comments raised in a previous round of review and you feel that this manuscript is now acceptable for publication, you may indicate that here to bypass the “Comments to the Author” section, enter your conflict of interest statement in the “Confidential to Editor” section, and submit your "Accept" recommendation.

Reviewer #2: All comments have been addressed

Reviewer #3: All comments have been addressed

2. Does this manuscript meet PLOS Global Public Health’s publication criteria? Is the manuscript technically sound, and do the data support the conclusions? The manuscript must describe methodologically and ethically rigorous research with conclusions that are appropriately drawn based on the data presented.

Reviewer #2: Yes

Reviewer #3: Yes

3. Has the statistical analysis been performed appropriately and rigorously?

Reviewer #2: N/A

Reviewer #3: Yes

4. Have the authors made all data underlying the findings in their manuscript fully available (please refer to the Data Availability Statement at the start of the manuscript PDF file)?

Reviewer #2: No

Reviewer #3: Yes

5. Is the manuscript presented in an intelligible fashion and written in standard English?

Reviewer #2: Yes

Reviewer #3: Yes

6. Review Comments to the Author

Reviewer #2: Thanks for addressing my comments! The manuscript has improved!

Best Wishes;

Labib

Reviewer #3: Thank you very much for this paper. This is a well written and rigorous study, well done. I noticed one minor issue that occurred twice "schools (x%)" - but I see no reason why this cannot be addressed while the paper is being copy-edited.

7. PLOS authors have the option to publish the peer review history of their article (what does this mean?). If published, this will include your full peer review and any attached files.

**Do you want your identity to be public for this peer review?** For information about this choice, including consent withdrawal, please see our Privacy Policy.

Reviewer #2: No

Reviewer #3: No
